# Vitamin D and Parathyroid Hormone during Growth Hormone Treatment

**DOI:** 10.3390/children9050725

**Published:** 2022-05-15

**Authors:** Teodoro Durá-Travé, Fidel Gallinas-Victoriano

**Affiliations:** 1Department of Pediatrics, School of Medicine, University of Navarra, Avenue Irunlarrea 1, 31008 Pamplona, Spain; 2Department of Pediatrics, Navarra Hospital Complex, 31008 Pamplona, Spain; fivictoriano@hotmail.com; 3Navarrabiomed (Biomedical Research Center), Health Department, Government of Navarra, 31008 Pamplona, Spain

**Keywords:** growth hormone deficiency, growth hormone treatment, parathyroid hormone, recombinant human growth hormone, vitamin D

## Abstract

**Background**. There is some controversy concerning a potential interaction between vitamin D and PTH and the GH/IGF-1 axis. The goal of this study is to assess vitamin D and PTH status in children with GH deficiency at diagnostic and during treatment with rhGH. **Methods**. Longitudinal and descriptive study in 110 patients, aged 3.3–9.1 years, with GH deficiency (GHD group) treated with rhGH. At diagnosis and after 12, 24, 36, and 48 months of treatment, a clinical (height, weight, and bone age) and laboratory (phosphorus, calcium, calcidiol, PTH, IGF-1) evaluation was performed. Concurrently, 377 healthy children, aged 3.8–9.7 years, were enrolled and constituted a control group. Vitamin D status was stated in accordance to the U.S. Endocrine Society criteria. **Results**. No significant differences were found in the prevalence of vitamin D deficiency among control (11.43%) and GHD (13.6%) groups at the moment of diagnosis, remaining without significant changes at 12 (12.9%), 24 (14.6%), 36 (13.1%), and 48 months (13.3%) of treatment. There were not any significant differences in serum levels of calcium, phosphorus, and calcidiol, but a steady increase (*p* < 0.001) in PTH was detected. **Conclusions**. Prepubertal patients with GH deficient do not appear to have a higher risk of vitamin D deficiency than healthy subjects, and with treatment with rhGH, no changes in the organic content of vitamin D were observed although a significant increase in PTH levels was detected.

## 1. Introduction

Changes in stature depend mainly on bone length growth, which is secondary, to a great extent, to the functionality of the growth plate in long bones. A sequential process of chondrocyte proliferation and hypertrophy takes place in the growth plate, along with the synthesis and mineralization of the extracellular matrix and vascular and osteoblastic invasion, where the plate is progressively replaced with bone tissue (endochondral ossification), thereby increasing in length [1].

Growth hormone (GH) and its major mediator IGF-1 (insulin-like growth factor), aside from its intense anabolic activity in the majority of body tissues, has a joint (endocrine–paracrine) action on the growth plate and, in this way, induces chondrocyte proliferation and differentiation and activates the mechanisms of osteoblastogenesis, thus contributing to length growth [2]. On the other hand, the process of endochondral ossification requires a precise calcium and phosphorous homeostasis, the preservation of which implies an adequate coordination among intestinal absorption, bone apposition, and resorption and renal excretion of both minerals, with vitamin D and parathyroid hormone (PTH) [3] as the main regulator hormones. In other words, vitamin D as well as PTH and the GH/IGF-1 axis would be essential in endochondral ossification and, consequently, in length growth. This fact justifies the interest aroused with regard to a hypothetical interaction between both hormone groups.

Several bibliographic reviews have been published in the last few years in relation to potential interactions among vitamin D and PTH and the GH/IGF-1 axis [4,5,6]. Even though it has been published, for instance, that vitamin D supplementation might increase IGF-1 or that GH as well as IGF-1 regulate renal calcidiol hydroxylation, the methodological heterogeneity makes data inconclusive.

The goal of this study is to determine the status of vitamin D and PTH in a group of prepubertal patients previously diagnosed with growth hormone deficiency at the time of diagnosis and throughout the period of treatment with rhGH. We pose the hypothesis that rhGH therapy may modify calcidiol and PTH serum levels in these patients.

## 2. Materials and Methods

### 2.1. Participants

This is a longitudinal descriptive study based on the retrospective analysis of the data collected in a convenience sample of 110 prepubertal children (46 boys and 64 girls) who had been previously diagnosed with isolated growth hormone deficiency, a so-called GHD group (maximum level of GH < 7.4 ng/mL after two different stimuli: clonidine and insulin). At the time of diagnosis, individuals ranged in age from 3.3 to 9.1 years, and all of them received treatment with recombinant human growth hormone (rhGH); they were subsequently followed up (clinical and analytical evaluation) every year for 48 months. All patients had weight and height adequate to gestational age at birth. Clinical evaluation was performed in the Pediatric Endocrinology Unit in our hospital in the period January 2010–December 2021.

Additionally, 377 prepubertal healthy individuals (159 boys and 218 girls) were recruited from the external consultations of other pediatric subspecialties, with an age range 3.8 to 9.9 years and weight and height adequate for sex and age (control group); they underwent clinical and analytical examination similar to that in the GHD group. All participants from GHD as well as control group who had received treatment with anticonvulsant drugs, glucocorticoids, or calcium or vitamin D supplementation in the period of 12 months prior to the inclusion in the study were excluded. Individuals who showed signs of pubertal development were also excluded; this means every patient that showed any sign of pubertal development during treatment with rhGH was automatically excluded beyond that point. In addition, no patient from the GHD group received any treatment with calcium of vitamin D supplements during the 48 months of treatment with rhGH.

### 2.2. Clinical Examination

Different clinical features (height, weight, BMI, and height velocity (HV)) were determined and collected at the onset of the treatment and 12 months afterwards for a period of four years of GH therapy. Participants were placed in underwear and barefoot for weight and height measurements. Weight measurement was performed using an Año-Sayol scale (reading interval 0–120 kg, precision of 100 g) and height measurement using a Holtain wall stadiometer (reading interval 60–210 cm, precision 0.1 cm). BMI was estimated by means of the corresponding formula: weight (kg)/height2 (m). Height, weight, BMI, and HV were presented as standard deviation scores (SDS). The SDS values for weight, height, BMI, and HV were calculated using the program Aplicación Nutricional from the Spanish Society of pediatric gastroenterology, hepatology, and nutrition (Sociedad Española de Gastroenterología, Hepatología, y Nutrición Pediátrica, available at https://www.gastroinf.es/nutritional/, accessed on 1 May 2021). The reference charts were taken from the study of Ferrández et al. (Centro Andrea Prader, Zaragoza 2002) [7]. Bone age (BA) was estimated using RUS-TW2 method [8]. 

### 2.3. Laboratory Determinations

A blood sample was collected (after overnight fasting) in all participants at the time of diagnosis and every 12 months for a period of 4 years after the onset of GH treatment in order to test for biochemical determinations (calcium, phosphorus, IGF-I, 25(0H)D, and PTH). The measurement of IGF-1 levels was completed with a solid-phase, enzyme-labeled chemiluminescent immunometric assay using an Immulite analyzer (Siemens, Erlanger, Germany). The quantification of calcium and phosphorus plasma levels was undertaken by colorimetric methods using a COBAS 8000 analyzer (Roche Diagnostic, Mannheim, Germany). A high-specific chemiluminescenceimmunoassay (LIAISON Assay, Diasorin, Dietzenbach, Germany) was used to measure 25(OH)D concentrations, and a highly specific solid-phase, two-site chemiluminescent enzyme-labeled immunometric assay in an Immulite analyzer (DPC Biermann, Bad Nauheim, Germany) was used to measure PTH concentrations. 

The individuals were classified in accordance to vitamin D plasma levels based on the criteria of the United States Endocrine Society. By doing so, a status of deficiency corresponded to 25(OH)D plasma levels lower than 20 ng/mL (<50 nmol/L), insufficiency to 25(OH)D levels between 20 and 29 ng/mL (50–75 nmol/L), and sufficiency to concentrations equal to or higher than 30 ng/mL (>75 nmol/L) [9].

### 2.4. Statistical Analysis

The results are presented as percentages (%) and means (M) and confidence intervals (95% CI). Chi-square, Kruskal–Wallis test and ANOVA (non-parametric test), and Pearson’s test when required were performed with the program Statistical Packages for the Social Sciences version 20.0 (Chicago, IL, USA). Statistical significance was considered when *p*-value was <0.05.

## 3. Results

In GHD group, at the time of diagnosis, the mean values obtained from blood draws were calcidiol: 29.38 ng/mL (CI 95%: 27.46–31.30); PTH: 31.90 ng/mL (CI 95%: 28.54–35.26); calcium: 9.89 mg/dL IC 95%: 9.81–9.97); and phosphorous 4.74 mg/dL (CI 95%: 4.62–4.85). In the control group, the mean values obtained were calcidiol: 28.28 ng/mL (IC 95%: 27.48–28.08); PTH: 29.54 (CI 95%: 28.29–30.79); calcium: 9.93 mg/dL (CI 95%: 9.89–9.97); and phosphorous: 4.64 mg/dL (CI 95%: 4.59–4.70). There were not any statistically significant differences in the mean values of these determinations between both groups. However, the mean IGF-1 values in the GHD group, 76.09 ng/mL (CI 95%: 67.86–84.33), were significantly lower (*p* < 0.001) than those in the control group: 145.55 ng/mL (IC 95%: 132.55–158.55).

The GHD group, at the time of diagnosis, displayed calcidiol levels above 30 ng/mL (vitamin D sufficiency) in 46.3% of the patients (n = 51), between 20–29 ng/mL (vitamin D insufficiency) in 40.9% (n = 44), and below 20 ng/mL (vitamin D deficiency) in 13.6% (n = 15). The analysis of the control group revealed that 44.3% (n = 167) of the participants showed vitamin D sufficiency, 44.3% (n = 167) insufficiency, and 11.4% (n = 43) deficiency. No statistically significant differences were found in the frequency of vitamin D status between the groups (chi-square: 1.86; *p* = 0.594).

Table 1 presents the mean values and the comparison of the different biochemical determinations between GHD and control group at the time of diagnosis based on the seasons of the year. In both groups, the lowest values for calcidiol corresponded to spring (GHD group: 24.41 ng/mL, control group: 25.68 ng/mL) and the highest values to summer (GHD group: 36.0 ng/mL, control group: 36.08 ng/mL); as for PTH, the lowest values corresponded to summer (GHD group: 25.88 pg/mL, control group: 26.08 pg/mL) and the highest values to autumn (GHD group: 33.75 pg/mL, control group: 32.17 pg/mL). No statistically significant differences were detected in plasma levels of calcium, phosphorous, and IGF-1 in the different seasons of the year between the groups. The comparison of both groups did not reveal any statistically significant difference in the mean values of calcium, phosphorous, calcidiol, and PTH in any season of the year except for the IGF-1 levels, which were significantly higher (*p* < 0.001) in the control group in comparison to the GHD in every season of the year.

Table 2 contains the clinical and biochemical features with mean values and the corresponding comparison for the DGH group at diagnosis as well as along the period of treatment with rhGH (at onset and after 12, 24, 36, and 48 months). A significant (*p* < 0.001) increase in mean values of weight-SD, height-SD and HV-DS, and PTH and IGF-1 levels was observed along the treatment period as well as a significant (*p* < 0.001) decrease in the mean values of the difference between chronological and bone age. There were not any statistically significant differences in the mean values of calcium, phosphorous, and calcidiol as well as in the dose of rhGH treatment in the different stages of follow-up.

Table 3 displays the mean values of calcium, phosphorous, calcidiol, and PTH during the treatment with rhGH and their comparison in accordance to the seasons of the year. Statistically significant differences (*p* < 0.001) were found in the mean values of calcidiol in every stage of follow-up, whose lowest levels corresponded to spring and the highest levels to summer; as for PTH, there were no significant differences in relation to the season of the year.

Table 4 displays the distribution of the prevalence of vitamin D status in every stage of follow-up along the treatment with rhGH (at onset and after 12, 24, 36, and 48 months). There were not any differences in the prevalence of the diverse vitamin D status among the different evaluations along the treatment with rhGH. In addition, no patient showed two consecutive episodes of vitamin D deficiency during the treatment with rhGH.

## 4. Discussion

The results obtained in this study emphasize the fact that vitamin D status (including seasonal variations) in prepubertal patients previously diagnosed with isolated growth hormone deficiency is similar to that in healthy participants (control group). In the same way, they let us note that these patients did not experience changes in vitamin D body content under the treatment with rhGH with respect to the moment of diagnosis even though a progressive and significant increase in PTH is observed during this period.

Growth in length of long bones depends, substantially, on the chondrocyte proliferation rate and the hypertrophy of the chondrocyte columns in the growth plate regulated by different hormones, particularly GH and IGF-1, as well as local factors (fibroblast growth factor, transforming factors, bone morphogenetic proteins, metaloproteinases, etc.) [10]. In fact, patients with GH deficiency suffer from abnormal bone remodeling, which would explain the failure in length growth and retardation in bone maturation (bone age) that is distinctive in these individuals; this event is progressively resolved after the onset of the treatment with rhGH (as we observed in this study). We should also note that the benefits from receiving the treatment with rhGH in the HGD group have been unequivocal during the analysed period, fulfilling largely the established effectiveness criteria [11,12,13]. The auxological response to the administration of rhGH has been quite satisfactory since the mean value for height-SD was −2.83 before treatment and reached −1.29 after 48 months of treatment, and simultaneously, IGF-1 levels increased significantly after initiating treatment with rhGH. The development and hypothesis proposed in this study required avoiding vitamin D supplementation. Taking into account the referred auxological response together with the fact that no patient displayed two chronologically consecutive episodes of vitamin D deficiency, we do not consider vitamin D supplementation mandatory in any case. 

The biological actions of vitamin D and PTH as regulators of mineral homeostasis prove to be complementary and contribute, on one side, to the growth in length of long bones by means of endochondral ossification and, on the other side, to the acquisition of the peak bone mass by means of a continuous positive balance of apposition/resorption. Vitamin D increases intestinal and proximal tubular calcium and phosphorous reabsorption [14] and boosts the mineralization of the bone extracellular matrix adjacent to the chondrocyte columns via the preservation of appropriate levels of these minerals. Even though in pathological conditions, such as hypovitaminosis D or hypocalcemia, PTH levels increase and exercise osteoclastic activity in order to obtain normal calcium plasma levels, in physiological conditions, it contributes to increase the bone formation rate, a fact that has given rise to considering PTH as a good marker for bone remodeling [15].

The mean values for calcium, phosphorous, calcidiol, and PTH in the GHD group at diagnosis were similar to those in the healthy individuals (control group), retaining the described seasonal variations [16,17,18]. However, even in the absence of significant changes in the mean values for calcium, phosphorous, and calcidiol along the treatment with rhGH, PTH levels remained significantly increased along the follow-up in comparison to the levels detected at diagnosis. Additionally, calcidiol seasonal variations remained constant in every follow-up evaluation along the treatment with rhGH, whereas PTH levels showed no seasonal changes. Therefore, the initial hypothesis in which treatment with rhGH seems to modify significantly PTH levels and its seasonal variations was, at least partially, confirmed. In this study, in accordance with other authors, a negative correlation between calcidiol and PTH plasma levels was detected in healthy individuals (control group) as well as in the participants of the DGH group at the moment of diagnosis, which is inherent of the physiological feedback mechanism between vitamin D and PTH secretion [16,17,18]. However, the lack of correlation between calcidiol and PTH plasma levels in the different evaluations along the period of treatment with rhGH together with a steady increase, with no seasonal variations, of PTH (not reaching levels of hyperparathyroidism) point out that the increase of PTH levels in these patients is secondary to a biological action of GH; the purpose of this action would be to boost the acquisition of bone mass and, consequently, growth in length. This eventuality leads to consider the possibility of an interaction between the regulatory hormones of mineral homeostasis (vitamin D and PTH) and the GH/IGF-1 axis in the skeletal development and growth in length and would justify the present interest in analyzing a potential functional interaction between both hormonal groups [4,5,6]. Nevertheless, published data regarding a hypothetical functional interaction between both groups in patients with GH deficiency are controversial since, for instance, references on the effects of the treatment with rhGH in vitamin D status vary from authors that refer a decrease [19] or no change [4] to those who refer an increase in vitamin D during the treatment with rhGH [20,21,22]. With respect to the published data in relation to the effects on PTH plasma levels, they consist of references relatively distant in time, and consequently, the highest sensibility of the present techniques of immunoassay could undermine the credibility of the results. This eventuality also tends to be contradictory since, when some experimental studies suggest that the axis GH/IGF-1 stimulates the activity of the renal 1-alpha-hydroxilase producing an increase in calcitriol synthesis and, consequently, increasing organic calcium and phosphorous availability and inhibiting PTH secretion [5,23], others do not appreciate any change in PTH during the treatment with rhGH [24]. However, in our case, we detected a significant increase in the levels of PTH that could be ascribed, as it has been previously considered, to an effect derived from the treatment with rhGH in order to enhance bone mass acquisition and, consequently, growth in length.

It should also be noted that the prevalence of vitamin D deficiency (calcidiol < 20 ng/mL) in the patients of the GHD group before the onset of the treatment with rhGH was similar to the healthy individuals (control group), whose growth was adequate for age and sex. This finding differs slightly from the previously published data from other authors that refer rates of vitamin D deficiency notably high (ranging from 40 to 44%) in this type of patients [20,21]. Furthermore, no significant differences in the seasonal variations of vitamin D levels between both groups were found, showing a higher vitamin D concentrations in the summer months, with a subsequent decrease in autumn and winter until the spring months, in which these concentrations reach a nadir [16,17,18]. In other words, the results we have obtained seem to indicate that there would not be a higher risk of vitamin D deficiency in those prepubertal individuals with GH deficiency with respect to the healthy population of the same age and that the content of vitamin D does not seem to be modified during the treatment with rhGH.

Concerning the revised literature, we emphasize that, in this study, in accordance with other authors [20,25,26], no correlation has been found either in healthy individuals (control group) or in patients of GHD group at diagnosis and during the treatment with rhGH between calcidiol levels and IGF-1 although some authors have described the referred correlation [22,27]. Nevertheless, different experimental studies suggest the existence of a complex functional paracrine/autocrine synergy between both elements on the growth plate [5,28]. Moreover, different authors have described, in patients diagnosed with nutritional rickets, how treatment with vitamin D involves an increase in IGF-1 levels with positive effects on length grow [29]. However, the majority of authors conclude considering that more prospective and randomized controlled trials are needed to understand the interaction between both hormonal groups and, in accordance to the conclusions subsequently drawn, to evaluate the possibility of a joint and general administration of vitamin D and rhGH to the patients with HGD or the supplementation with vitamin D exclusively in those patients in deficiency.

All the participants included in this study, in the control group as well as in the GHD group, presented in a prepubertal stage, with those who showed signs of puberty being deliberately excluded previously. This option was motivated mainly because of, on one side, the intense activity of the growth plate during puberty secondary to the action of the sexual hormones and, on the other side, because it has been proved that there is a higher risk of hypovitaminosis D during puberty [15,16], and both eventualities may act as confounding factors.

One limitation of our study is that we do not have simultaneous biochemical data from the control group and the GHD group (at 12, 24, 36, and 48 months of follow-up). This could have allowed assessing the potential influence of age on the PTH levels. However, the results obtained are explicit enough to mitigate this limitation. 

## 5. Conclusions

In essence, the results of this study show that prepubertal patients with GH deficiency do not seem to have a higher risk to present with vitamin D deficiency than healthy individuals of the same age. In addition, rhGH treatment does not modify vitamin D body content, including the seasonal variations, even though a significant increase in PTH levels has been observed with no seasonal variations. In any way, more prospective and randomized controlled trials are required for a better understanding of a hypothetical interaction between vitamin D/PTH and the GH/IGF-1 axis.

## Figures and Tables

**Table 1 children-09-00725-t001:** Baseline biochemical characteristics in accordance to the season between GHD and control groups (M, 95% CI).

GHD Group (n = 110)	Winter (n = 24)	Spring (n = 29)	Summer (n = 26)	Autumn (n = 31)	(*p*) *
Calcium	9.95 (9.76–10.14)	9.78 (95.6–9.99)	9.99 (9.91–10.06)	9.88 (9.76–10.00)	0.260
Phosphorous	4.74 (4.53–4.95)	4.70 (4.42–4.99)	4.76 (4.58–4.94)	4.79 (4.59–4.99)	0.318
Calcidiol	27.20 (23.06–30.79)	24.41 (20.46–28.36)	36.00 (32.43–39.56)	28.00 (24.73–31.26)	0.001
PTH	30.77 (25.77–35.77)	31.66 (26.43–36.90)	25.88 (18.96–32.79)	33.75 (24.40–43.09)	0.022
IGF-1 **	82.41 (61.45–103.36)	74.37 (61.66–87.08)	66.40 (49.54–83.27)	82.24 (62.26–102.22)	0.502
Control group (n = 377)	Winter (n = 112)	Spring (n = 85)	Summer (n = 78)	Autumn (n = 102)	ANOVA (*p*)
Calcium	9.94 (9.88–10.00)	9.90 (9.81–9.99)	10.0 (9.95–10.07)	9.89 (9.83–9.96)	0.179
Phosphorous	4.65 (4.56–4.74)	4.54 (4.40–4.68)	4.72 (4.58–4.85)	4.66 (4.56–4.76)	0.320
Calcidiol	26.55 (25.31–27.78)	25.69 (23.86–27.52)	36.08 (33.78–38.39)	28.2 (26.96–29.54)	0.001
PTH	29.92 (27.78–32.06)	31.88 (28.94–34.82)	26.08 (23.52–28.63)	32.17 (29.83–34.51)	0.017
IGF-1 **	145.56 (132.99–158.13)	148.33 (135.43–161.23)	151.71 (138.49–164.93)	143.44 (131.29–155.59)	0.677

* Kruskal–Wallis test. ** Student’s *t*-test, *p* < 0.05 between groups.

**Table 2 children-09-00725-t002:** Clinical and biochemical features in the GHD group at diagnosis and along the treatment with rhGH (M, 95% CI).

DGH Group	Time of Follow-Up					ANOVA (*p*)
	At Onset (n = 110)	After 12 m (n = 108)	After 24 m (n = 109)	After 36 m (n = 106)	After 48 m (n = 90)	
Age (years)	6.32 (6.01–6.63	7.30 (7.06–7.54)	8.36 (8.11–8.61)	9.16 (8.87–9.45)	10.23 (9.81–10.69)	0.001
Weight-SD	−1.52 (−1.62/−1.41)	−1.26 (−1.36/−1.16)	−1.13 (−1.24/−1.03)	−0.95 (−1.09/−0.80)	−1.00 (−1.14/−0.86)	0.001
Height-SD	−2.83 (−2.95/−2.71)	−1.98 (−2.12/−1.84)	−1.73 (−1.86/−1.60)	−1.36 (1.52/−1.20)	−1.29 (−1.49/−1.09)-	0.001
BMI-SD	−0.49 (−0.64/−0.35)	−0.67 (−0.76/−0.57)	−0.60 (−0.78/−0.54)	−0.67 (−0.80/−0.54)	−0.69 (−0.85/−0.52)	0.001
Dose (mcg/kg/d)	---	33.46 (32.95–33.98)	31.26 (30.66–31.86)	31.64 (30.78–32.50)	31.91 (31.00–32.82)	0.681
HV-SD	−2.45 (−2.64/−2.26)	4.34 (3.93–4.75)	1.69 (1.40–1.97)	1.59 (1.18–2.00)	1.58 (0.73–2.22)	0.001
CA/BA	1.24 (1.16–1.33)	1.19 (1.13–1.25)	1.11 (1.07–1.15)	1.05 (1.01–1.08)	1.03 (1.00–1.06)	0.001
Calcium (mg/dL)	9.89 (9.81–9.97)	9.84 (9.78–9.89)	9.78 (9.69–9.87)	9.77 (9.70–9.84)	9.82 (9.69–9.95)	0.653
Phosphorous (mg/dL)	4.74 (4.62–4.85)	4.79 (4.71–4.87)	4.80 (4.70–4.90)	4.76 (4.67–4.85)	4.81 (4.68–4.94)	0.302
Calcidiol (ng/mL)	29.38 (27.46–31.30)	28.03 (26.80–29.27)	27.38 (26.15–28.61)	28.74 (26.82–30.66)	28.36 (26.59–30.13)	0.501
PTH (pg/mL)	31.9 (28.54–35.26)	37.39 (35.36-.40.21)	37.76 (35.09–40.42)	39.77 (37.09–42.46)	40.83 (37.08–44.57)	0.008
IGF-1 (ug/mL)	76.09 (67.86–84.33)	199.53 (185.19–213.86)	212.82 (197.83–227.82)	215.50 (199.33–231.68)	223.10 (204.40–241.79)	0.001

HV, height velocity; CA/BA, chronological/bone age ratio. m: months.

**Table 3 children-09-00725-t003:** Biochemical features in accordance to the season of the year in the GHD group along the treatment with rhGH (M, 95% CI).

Follow-Up	Season of the Year				*p* *
	Winter	Spring	Summer	Autumn	
After 12 m	(n = 29)	(n = 26)	(n = 26)	(n = 27)	
Ca (mg/dL)	9.86 (9.67–10.06)	9.85 (9.76–9.94)	9.95 (9.85–10.05)	9.71 (9.60–9.82)	0.370
P (mg/dL)	5.23 (5.00–5.45)	5.05 (4.92–5.17)	5.03 (4.84–5.21)	5.08 (4.93–5.23)	0.412
Calcidiol (ng/mL)	27.90 (26.26–29.53)	24.16 (21.43–26.89)	33.35 (30.18–36.52)	25.94 (23.58–28.30)	0.001
PTH (pg/mL)	35.34 (32.13–38.56)	42.10 (35.87–48.32)	38.00 (32.11–43.88)	38.77 (32.88–44.66)	0.315
After 24 m	(n = 32)	(n = 24)	(n = 26)	(n = 28)	
Ca (mg/dL)	9.83 (9.67–9.99)	9.77 (9.64–9.90)	10.06 (9.84–10.27)	9.65 (9.44–9.87)	0.890
P (mg/dL)	5.03 (4.02–5.24)	4.94 (4.76–5.12)	4.95 (4.70–5.20)	4.91 (4.77–5.05)	0.575
Calcidiol (ng/mL)	26.90 (25.05–28.76)	21.16 (19.32–23.00)	31.77 (28.25–35.29)	27.60 (25.28–29.91)	0.001
PTH (pg/mL)	37.55 (33.61–41.49)	34.60 (30.14–39.05)	38.11 (20.93–45.29)	43.46 (37.19–49.73)	0.661
After 36 m	(n = 34)	(n = 22)	(n = 26)	(n = 24)	
Ca (mg/dL)	9.83 (9.67–9.98)	9.73 (9.54–9.92)	9.74 (9.59–9.89)	9.68 (9.55–9.80)	0.515
P (mg/dL)	4.74 (4.57–4.91)	4.83 (4.56–5.10)	4.76 (4.66–4.86)	4.81 (4.59–5.02)	0.432
Calcidiol (ng/mL)	27.58 (22.71–32.44)	22.00 (20.25–23.74)	37.84 (34.37–41.31)	27.91 (25.52–30.30)	0.001
PTH (pg/mL)	41.00 (36.44–45.55)	38.26 (33.22–43.31)	41.66 (35.68–47.65)	39.00 (33.08–44.91)	0.772
After 48 m	(n = 25)	(n = 23)	(n = 22)	(n = 20)	
Ca (mg/dL)	9.55 (9.37–9.73)	9.63 (9.28–9.98)	9.58 (9.44–9.72)	9.86 (9.59–10.13)	0.409
P (mg/dL)	4.76 (4.59–4.94)	4.85 (4.48–5.22)	4.75 (4.56–4.93)	4.90 (4.46–5.33)	0.513
Calcidiol (ng/mL)	22.83 (20.03–25.63)	21.14 (17.83–24.44)	31.16 (28.52–33.80)	30.40 (28.28–32.51)	0.001
PTH (pg/mL)	42.36 (35.93–48.79)	41.00 (33.84–48.15)	38.00 (29.55–46.45)	39.66 (32.21–47.11)	0.848

* Kruskal–Wallis test.

**Table 4 children-09-00725-t004:** Prevalence of calcidiol status in the different stages of follow-up in the GHD group.

DGH Group (Follow-Up)	Vitamin D Deficiency	Vitamin D Insufficiency	Vitamin D Sufficiency
At onset (n = 110)	15 (13.6%)	44 (40.9%)	51 (46.3)
After 12 m (n = 108)	14 (12.9%)	49 (45.4%)	45 (41.7%)
After 24 m(n = 109)	16 (14.6%)	51 (46.7%)	46 (42.2%)
After 36 m (n = 101)	14 (13.8%)	42 (41.5%)	44 (43.5%)
After 48 m (n = 90)	12 (13.3%)	41 (45.5%)	37 (41.1%)

Chi-square: 8.909 (*p* = 0.431).

## Data Availability

All data generated or analyzed during this study are included in this article. Further inquiries can be directed to the corresponding author.

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
