# Peer review of "Vitamin D and Parathyroid Hormone during Growth Hormone Treatment"

_children, 2022, doi:10.3390/children9050725_

Round 1
Reviewer 1 Report
It seems strange to me that you have printed and given a doi number before the Ms has been reviewed.
This is a well written paper, indicating the Vit D levels in children in North Spain, in controls and GH deficient. There are 2 major points:
Vit D levels differ in various geographical regions thus the data is geographical area bound.
The fact that Vit D is not related to GH is not a novelty.
The statement that these findings are remarkable should be deleted.
Reviewer 2 Report
Dear Authors,
Your manuscript entitled “Vitamin D and parathyroid hormone during growth hormone treatment” offers a very interesting description of the effects of this therapy in children.
The study is plenty of remarkable values:
- The long period of follow-up
- The existence of a comparable control group.
- The analysis of variations according to the season of the year.
- The discussion is brilliant.
I have only one question: why did you decide to not supplement with Vitamin D in cases of deficiency? I believe that this decision should be explained in detail at the discussion.
Author Response
Response to Reviewer 2 Comments
We submit the corrections of the different aspects you considered in relation to the manuscript entitled “Vitamin D and parathyroid hormone during growth hormone treatment” (children-1662153).
According to the suggestions of the reviewer, the following modifications have been made:
1) Your manuscript entitled “Vitamin D and parathyroid hormone during growth hormone treatment” offers a very interesting description of the effects of this therapy in children.
- The study is plenty of remarkable values:
- The long period of follow-up
- The existence of a comparable control group.
- The analysis of variations according to the season of the year.
- The discussion is brilliant.
We would like to express our gratitude for the words of the reviewer. They motivate us to continue making an effort in our daily clinical and research work.
2) I have only one question: why did you decide to not supplement with Vitamin D in cases of deficiency? I believe that this decision should be explained in detail at the discussion.
In Results (table 4), the following sentence has been added:
“In addition, no patient showed two consecutive episodes of vitamin D deficiency during the treatment with rhGH”
In Discussion ( 2nd paragraph), the following sentences have been added:
“The aim and hypothesis proposed in this study required avoiding vitamin D supplementation. Taking into account the referred auxological response together with the fact that no patient displayed two chronologically consecutive episodes of vitamin D deficiency, we do not consider vitamin D supplementation mandatory in any case”.
We would like to express our thanks to referee for your suggestions and positive criticisms.
We hope every made question have been answered adequately.
Yours sincerely,
Teodoro Durá-Travé